# The effects of assessing character strengths vs. psychopathology on mood, hope, perceived stigma and cognitive performance in individuals with psychosis

**Aman Randhawa**[1]*, **Simone Kühn**[1,2], **Daniel Schöttle**[1], **Steffen Moritz**[1], **Jürgen Gallinat**[1], **Leonie Ascone**[1]

**1** Universitätsklinikum Hamburg-Eppendorf, Clinic and Policlinic for Psychiatry and Psychotherapy, Hamburg, Germany, **2** Lise-Meitner Group for Environmental Neuroscience, Max Planck Institute for Human Development, Berlin, Germany

* Aman.r@outlook.de

**Data Availability Statement:** The data underlying the results presented in the study are available

## Abstract

### Objectives

The main objective of the present study was to investigate whether assessments of psychopathology vs. character strengths were associated with systematic differences concerning transient psychological states (i.e., cognitive performance, state mood, optimism, therapy motivation, perceived stigma) in individuals with psychotic disorders. An additional goal was to evaluate the acceptance and appraisal of a subsequent online character-strength intervention, consisting of top-two strengths feedback, and to explore associations between character strengths and psychotic symptoms. The study thus aimed to contribute to the discussion on the extension of current treatment approaches for schizophrenia through positive psychological interventions.

### Methods

The study was implemented online applying a randomized within-subject cross-over design in N = 39 patients with self-reported psychosis. After a baseline assessment, briefly capturing psychological states (including cognition: TMT A/B, positive and negative affect, motivation for change/ therapy, optimism, and self-stigma) participants were randomly assigned to a first questionnaire block, which addressed either individual character strengths (VIA-IS) or psychopathology (CAPE & BSI). This was followed by a second, brief assessment of transient psychological states, whereafter the second questionnaire block was conducted, this time with the respective opposite (strengths or psychopathology) assessment. A final psychological states assessment was conducted. Afterwards, participants received feedback on their top-two strengths and a brief psycho-education, followed by a qualitative assessment.

from the Harvard Dataverse Repository (https://doi.org/10.7910/DVN/1ZHNO1).

**Funding:** The authors received no specific funding for this work.

**Competing interests:** The authors have declared that no competing interests exist.

## Results

Contrary to expectations, there were no differences between the psychological states after the pathology vs. character strengths assessment blocks. Character strengths mainly correlated negatively with negative symptoms, with medium to large effect sizes. Participants were generally satisfied with the intervention and rated a focus on personal strengths in psychotherapy as highly important.

## Conclusion

Our main hypothesis stating that the assessment of character strengths (vs. psychopathology) is associated with differences in subsequent psychological states could not be confirmed. Qualitative findings indicate that the emphasis on individual character strengths interventions is well accepted and viewed as important. The associations of character strengths with negative symptoms are important from the background of the cognitive model or defeatist beliefs (e.g., amotivation due to perceiving the self as 'incapable'), which could be addressed in experimental or intervention studies targeting character strengths.

## Introduction

Traditionally, psychiatry has been defined and established as a profession and research domain that focuses on the study and treatment of mechanisms of mental pathology, including psychosis. However, the field is experiencing a current expansion, increasingly incorporating an additional emphasis on outcomes which had formerly often been—and still are—labelled as 'secondary', such as e.g., well-being, coping, self-esteem, or social functioning. This broadening of perspective has significantly been promoted by the 'recovery movement', which has been criticizing the deficit- and often deterministic view on individuals affected by (particularly severe) mental illness [1, 2]. A paradigm shift is now becoming more and more visible. For instance, it has been proclaimed that the time of 'positive psychiatry' has come [3], advocating a more holistic view whereby the focus on psychopathology is necessarily complemented by studying, emphasizing, and fostering personal strengths and resources, as well as positive mental health outcomes (e.g. well-being, positive emotions), and psychological traits related to resilience (e.g., optimism, compassion, spirituality).

Quite similarly, positive psychology has explicitly been intended as a balancing, alternative view and research agenda to complement the deficit-focused DSM [4]. This field of research is dedicated to uncovering the mechanisms and means for the realization of positive human qualities, values, and emotions. Thereby, 'meaning' as a source of resilience, motivation and endurance plays a pivotal role. Peterson and Seligman, who significantly contributed to establishing positive psychology as a research field, identified so-called character strengths (e.g., courage, temperance, love of knowledge or wisdom, creativity), as a source of positive self-identity and meaning, comprising traits considered as desirable across different cultures [5]. Interventions that address learning about personal character strengths and how to make use of them have shown to have a beneficial effect on individuals, such as alleviating depressive symptoms or enhancing life satisfaction and positive emotions [6].

On the other hand, the impact of negative views upon the self, or self- / social stigma, can have detrimental effects on different levels, including mental health and functioning of an individual or certain, stigmatized groups. Recent work implies that the role of (biologically determined) neurocognitive deficits in the level of functioning in individuals with schizophrenia

may have been overemphasized and overestimated. It has been shown that neurocognitive performance is significantly influenced by attitudinal and motivational factors such as dysfunctional beliefs, avolition, and negative emotions. Beck et al. [7] propose a cognitive model based on dysfunctional beliefs that provides a comprehensive approach to explain poor performance in laboratory tasks and poor level of functioning among individuals with schizophrenia, challenging the assumption that there is a neurologically-based neurocognitive deficit in schizophrenia that is independent of other psychological processes. The core component of the therapy derived from this model, labeled as recovery-oriented cognitive therapy (CT-R), is to strengthen positive beliefs and weaken negative beliefs. Moritz et al. [8] come to the same conclusion that the neurocognitive deficits in schizophrenia might have been overestimated, among other factors partially due to motivational effects, stereotypes or even stigma (i.e., presupposing cognitive deficits), or defeatists beliefs, and that taking these factors into account opens new opportunities for treatment.

Albeit the expansion of research in psychiatry and psychology towards focusing more on positive human traits and emotions seems both logical and broadly appreciated, the indication and benefits of related interventions need to be the object of well-controlled investigation. Thus far to our knowledge, there have only been a handful of studies addressing character strengths in psychosis [9, 10]. The study by Browne et al. was a post-hoc analysis of data from a multimodal psychotherapy treatment program (NAVIGATE trial) which showed that character strengths prospectively correlated with lower symptom levels of psychosis. However, the study used a customized version of the character strengths questionnaire, with unclear reliability. The study by Sims et al. had a small sample size (N = 29) and no control condition, which makes the identified positive effects of a character strength identification intervention on elevated positive, reduced negative mood, and improved cognitive performance (Trail Making Test) hard to interpret.

Given the findings on the role of dysfunctional beliefs in neurocognitive performance and level of functioning in schizophrenia, the potential beneficial effects of directing attention towards positive self-attributes, and the currently limited research on addressing character strengths in psychosis, we conducted this study to contribute to the discussion on whether focusing on character strengths could be an important contribution to alleviate the stereotypical, deficit-oriented view on individuals with schizophrenia and also their view of themselves. In the long run, working with character strengths might then improve symptomatic and functional outcomes. To begin, we took one step back and investigated whether merely measuring psychopathology vs. character strengths may already have an effect on psychological states. In addition, the present study aimed at exploring correlations between character strengths and psychotic symptoms to potentially identify specific symptom domains for which addressing character strengths could be particularly fruitful in treatment. Finally, the general acceptance of focusing on character strengths was investigated.

## Methods

### Procedure

The study used various recruitment methods, including flyers and social media, as well as a mailing list of former study participants from the Medical Center Hamburg-Eppendorf (UKE). Inclusion criteria were age > 18 years, self-reported formal diagnosis of a psychotic disorder, sufficient emotional stability, and being able and willing to give informed consent after reading the study information. The study was conducted online and designed as a randomized-controlled within-subject cross-over study.

Once participants provided informed consent, they were assessed for cognitive function using the Trail Making Test A and B (TMT A&B), as well as psychological states (mood, optimism, therapy motivation, and perceived stigma), which served as the baseline measures.

Participants were then randomly assigned to either a psychopathology assessment block or a character strengths assessment block. After completing the assigned assessment block, participants underwent another round of assessment of the TMT A&B and psychological states. They then completed the respective other assessment block (psychopathology or character strengths), followed by a final assessment of the TMT A&B and psychological states.

After that, all participants received a brief character strength intervention, which consisted of personalized feedback on their top-two character strengths, as well as an instruction sheet with easy-to-implement exercises to make use of their character strengths in everyday life.

At the end of the study, socio-demographic information, including current or past diagnoses and therapy status, were assessed. To distinguish potential 'simulators' from real patients in this online study, a 4-item psychosis lie scale (cutoff: 8 points); [11] was used ($\alpha$ = .62), which was integrated within the CAPE (see Psychopathology subsection for details), whereby the respective items reflect clichés about psychosis, but are, in fact, rather uncommon (e.g., alien abduction). Scores beyond cutoff speak for simulation of psychosis. This study, including all conducted analyses, was conducted with the formal approval of the ethics committee of the University Medical Center Hamburg-Eppendorf (LPEK-0166, approval date 06.07.2020).

## Measures

**Cognitive test: Trail-Making-Test A&B.**   A computerized version of the original TMT [12] developed by Timo Gnambs for Unipark (EFS Survey; Questback, https://timo.gnambs. at/research/tmt), was used. After a practice trial, in version A the participants had to connect the numbers 1 to 25 as fast as possible in ascending order using the computer mouse. In version B, numbers and letters had to be connected alternately and in ascending order, from A to L and 1 to 13. The seconds needed to correctly complete TMT A and B were shown to the participant, and he or she had to type it into a text field. The central outcome parameter in this study was TMT-B minus TMT-A (TMTB-A). The TMT is a frequently used indicator of executive functions (i.e., processing speed, task-set inhibition, flexibility, attention). The difference reflects the degree by which the time to complete the task increases by introducing the shifting component.

**Psychological states.**   To assess state affect, the Positive and Negative affect schedule [13] was used ($\alpha_{positive}$: .86; $\alpha_{negative}$ = .76). The scale comprises 10 affective adjectives each, describing positive or negative states, which are rated in a 5-point Likert scale of how strongly each state applies at the moment (1 = 'very slightly or not at all' to 5 = 'extremely'). To assess the current level of motivation for therapy or change in general, the University of Rhode Island Change Assessment Scale [14] was slightly modified and state adapted for the purpose of the present study ($\alpha$ = .79). It comprises 9 items, rated on a 5-point Likert, ranging from 1 = 'does not apply to me' to 5 = 'fully applies to me'. The state optimism measure [15], which consists of 7 items and applies a 5-point-Likert scale format (1 = 'do not agree' to 5 = 'fully agree') was used ($\alpha$ = .97). Perceived stigma was assessed as defined in the World Mental Health surveys: reporting perceived (mental) health-related embarrassment or shame and discrimination [16] in a state-adapted short form, using two items from the original scale ('*How strongly do you feel discriminated or unfairly treated due to your health problems*?' and '*How strongly do you feel ashamed due to your health problems*?'; rated from 1 = 'not at all' to 5 = 'extremely'), ($\alpha$ = .92).

**Character strengths.** Character strengths were measured with the 120 items short version of the Values in Action Inventory of Strengths [17] in a validated German version [18]. In total, 24 character strengths are assessed with 5 items per strength. The 24 strengths in turn are assigned to 6 superordinate human virtues: *wisdom and knowledge* (strengths: creativity, curiosity, judgment, love of learning, perspective), *courage* (strengths: bravery, perseverance, honesty, zest), *humanity* (strengths: love, kindness, social intelligence), *justice* (strengths: teamwork, fairness, leadership), *temperance* (strengths: forgiveness, humility, prudence, self-regulation) and *transcendence* (strengths: appreciation of beauty and excellence, gratitude, hope, humor, spirituality), although the factorial structure of the questionnaire has been challenged in later factor analyses [19]. Each item describes a respective strength (e.g., creativity: '*I am always coming up with new ways to do things*') and is rated on a 5-point Likert scale (1 = 'very much unlike me' to 5 = 'very much like me'), (VIA-IS scales: $\alpha_{median}$ = .72; $\alpha_{range}$ = .33 - .90).

**Psychopathology.** To match the number of items focusing on character strengths (120), the Brief Symptom Inventory [20] (53 items) and the Community Assessment of Psychic Experiences [21] (with 42 items on the frequency of positive, negative and depressive symptoms; if reported frequency > never, this is followed by a symptom-related distress assessment, maximum of 84 items) were used. Both questionnaires referred to the last 7 days. The BSI (total $\alpha$ = .94) assesses nine symptom dimensions (somatization, obsessive-compulsive symptoms, hostility, interpersonal sensitivity, depression, general anxiety, phobic anxiety, paranoid ideation, psychoticism) as well as bad appetite, sleep difficulties, suicidal ideation, and guilt, whereby all items are rated on a 5-point Likert scale (0 = 'not at all' to 4 = 'extremely'). The CAPE ($\alpha_{positive}$ = .90; $\alpha_{negative}$ = .88; $\alpha_{depressive}$ = .81) is usually evaluated concerning the three subscales of positive symptoms (20 items; symptom dimensions: bizarre experiences, hallucinations, paranoia, grandiosity, magical thinking), negative symptoms (14 item; symptom dimensions: social withdrawal, affective flattening, avolition) and depressive symptoms (8 items). All items are first rated on a frequency scale concerning their occurrence in a certain period of time (options ranging from 1 = 'never' to 4 = '(nearly) all of the time'). If an item is rated > 1, the distress evoked by the respective symptom is rated on another 4-point Likert scale (1 = 'not at all distressing' to 4 = 'very distressing').

**Character strengths intervention and qualitative assessments.** The intervention consisted of targeted feedback concerning personalized top-two character strengths (those with the highest score, as computed automatically by the program), informing the participant about the definition and meaning of these strengths, and a short psycho-education about the benefits of identifying and applying one's strengths in everyday life, whereby presenting easy-to-implement mini-interventions. The latter were selected from a repertoire of original instructions on how to work with character strengths from the VIA-Institute of Character (https://www.viacharacter.org/) and adapted to be suited for the present study. Participants could get an entire character strengths profile on demand via providing an anonymous e-mail-address at the end of the study.

The qualitative assessment consisted of a customized version of the ZUF-8 [22], a widely used tool to assess patient satisfaction, with the exception of items that addressed satisfaction with the treatment institution/ clinic, as this did not apply at all to the context of the present study. The remaining items addressed the perceived quality of the intervention, met vs. unmet expectations, usefulness for personal problems, recommendation to a friend, and general contentment (4-point Likert scale rating format; e.g., 1 = 'bad/ not satisfied' to 4 = 'excellent/ very satisfied'), ($\alpha$ = .93). In addition, three study-specific items were designed, asking participants about: the identification/ satisfaction with the feedback on personal top-strengths (dichotomous; yes vs. no), and the perceived importance to address character strengths on a scale from 1 = 'not important' to 5 = 'very important'.

## Statistical analyses

A series of paired t-tests was carried out to compare whether there were any differences in mood (positive and negative affect), optimism, therapy motivation, and perceived stigma a) between baseline vs. after the assessment block of character strengths, b) baseline vs. after the assessment block of psychopathology, and c) between the two assessment blocks of character strengths vs. psychopathology. No alpha-level adjustments were made due to the exploratory nature and novelty of this study. If the assumptions for parametric paired t-tests were not met, the non-parametric Wilcoxon signed rank test was conducted. Non-parametric correlation analyses were carried out to check associations between psychotic symptom domains (frequency dimension) and character strengths. Qualitative data was evaluated descriptively.

# Results

## Sample characteristics and descriptive statistics

A total of $N = 53$ participants took part in the study, of which four terminated the study at the beginning (1 declined consent, 3 participants had heightened suicidality and were subsequently led to the suicidality help page; see Methods section). Furthermore, of the remaining 49 participants, 10 reported to never have experienced psychosis, which is why these cases were excluded from further analysis. Mean age of the remaining sample ($N = 39$) was 43 years ($SD = 9.9$). Four (10.3%) individuals reported past episodes of psychosis, 27 (69.2%) reported both past and a current episode(s), and 8 (20.5%) reported only a current episode. Self-reported number of episodes ($N = 37$; one case was dismissed with 111 reported episodes—possible entry error; one participant stated that he/ she stopped counting as having had too many episodes—set to missing) ranged between 1 and 25 (*Median* = 5.00, $SD = 5.80$). Mean age of onset was 24 years ($SD = 6.23$, range: 15 to 43 years). 66.7% of the sample reported having the highest German school degree. Of the total sample, 20.5% were working (7.7% full-time, 12.8% part time), 10.3% were students, and 33.3% were in early retirement (disability pension). The remaining 35.9% of participants indicated various statuses, such as active job search, mini-jobs, participation in rehabilitation programs, looking after home or children, etc. Participants had a mean frequency score (item score) of 1.44 ($SD = 0.47$) on the CAPE positive [POS], 2.55 ($SD = 0.45$) on the negative [NEG] and 1.86 on the depression [DEP] scale ($SD = 0.56$). Note that these values are similar to the ones reported in the Genetic Risk and Outcome of Psychosis Project study (n = 868 individuals with a psychotic disorder; POS mean = 1.68, SD = 0.50; NEG mean = 2.02, SD = 0.53; DEP = 2.00, SD = 0.58), [23]. None of the participants scored above cutoff on the psychosis lie scale (see Methods section for details).

## Effects on cognitive performance and psychological states

There were no differences in the psychological states ratings or cognitive performance between the pathology vs. character strengths assessments. Time effects were observed, with the variables TMT, negative affect and stigma decreasing in both conditions compared to baseline. For statistical details, see Table 1.

## Descriptive data on character strengths and correlation analyses

Kindness, honesty and judgment were the three descriptively most highly endorsed character strengths; the three least endorsed character strengths were self-regulation, spirituality, and humility. For a full average profile, see Fig 1.

Ten of the 24 character strengths were significantly and negatively correlated with negative symptoms (see Table 2). Positive correlations with positive symptoms were identified for two

**Table 1. Descriptives and within-sample t-test results concerning outcomes of interest.**

| Variable | M (SD) Baseline[a] | M (SD) Pathology[b] | Mean (SD) Strengths[c] | Inferentials | Effect size[*2] |
|---|---|---|---|---|---|
| TMT(B-A)[*1] | 22.9 (24.0) | 14.8 (8.1) | 14.2 (7.9) | $z_{a,b} = -2.69, p = .007$ | -0.08 |
| | | | | $z_{a,c} = -3.27, p = .001$ | -0.08 |
| | | | | $z_{b,c} = -0.04, p = .967$ | -0.01 |
| Positive affect | 30.3 (6.9) | 30.2 (7.5) | 30.0 (7.6) | $t_{a,b} = 0.18, p = .859$ | 0.02 |
| | | | | $t_{a,c} = 0.38, p = .709$ | 0.04 |
| | | | | $t_{b,c} = 0.21, p = .832$ | 0.02 |
| Negative affect[*1] | 17.0 (4.5) | 14.7 (4.5) | 14.6 (5.0) | $z_{a,b} = -3.25, p = .001$ | -0.08 |
| | | | | $z_{a,c} = -3.45, p = .001$ | -0.09 |
| | | | | $z_{b,c} = -0.90, p = .370$ | -0.02 |
| Motivation[*1] | 34.5 (6.8) | 35.2 (7.6) | 35.0 (7.8) | $z_{a,b} = -0.79, p = .430$ | -0.02 |
| | | | | $z_{a,c} = -1.00, p = .317$ | -0.03 |
| | | | | $z_{b,c} = -0.47, p = .637$ | -0.01 |
| Optimism[*1] | 23.7 (5.8) | 23.9 (6.5) | 23.7 (6.5) | $z_{a,b} = -0.66, p = .513$ | -0.02 |
| | | | | $z_{a,c} = -0.60, p = .551$ | -0.02 |
| | | | | $z_{b,c} = -0.06, p = .956$ | -0.00 |
| Stigma[*1] | 4.97 (2.15) | 4.23 (2.08) | 4.23 (2.02) | $z_{a,b} = -3.44, p = .001$ | -0.09 |
| | | | | $z_{a,c} = -3.68, p < .001$ | -0.09 |
| | | | | $z_{b,c} = 0.00, p = 1.00$ | 0.00 |

**Note.**

[*1] non-normal difference score distribution, hence application of the non-parametric Wilcoxon signed rank test.

[*2] Effect sizes were *r* for non-parametric analyses and Cohen's *d* for parametric analyses.

character strengths (fairness, leadership). For depression there were four negative correlations and one positive correlation with character strengths (for details see Table 2).

## Acceptance of the character strengths intervention

The median satisfaction rating was 3.10 (SD = 0.74), indicating an overall good quality and satisfaction with the character strengths intervention (scale range: min. = 1; max. = 4). Overall, 97.4% (n = 38) of the participants could identify with their top-two character strengths, one

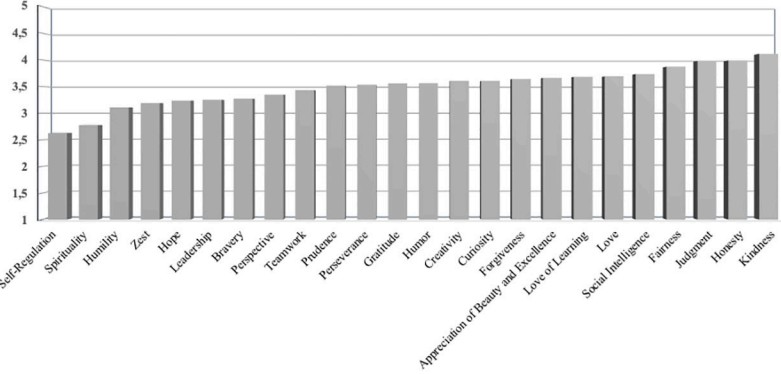

**Fig 1. Profile of character strengths endorsement by individuals with a self-reported psychotic disorder.**

**Table 2. Spearman-rho correlations of character strengths with psychotic symptoms and depression CAPE mean frequency scores ($N = 39$).**

| Domain | Character strength | Positive symptoms | Negative symptoms | Depression |
|---|---|---|---|---|
| **Wisdom & Knowledge** | Creativity | .092 | -.308[†] | -.055 |
| | Curiosity | .149 | **-.555***** | -.201 |
| | Judgment | -.114 | .230 | .219 |
| | Love of Learning | .170 | -.289[†] | .066 |
| | Perspective | .037 | -.129 | -.005 |
| **Courage** | Bravery | .154 | **-.370*** | -.076 |
| | Perseverance | .083 | **-.343*** | -.082 |
| | Honesty | .171 | -.042 | .119 |
| | Zest | .087 | **-.755***** | **-.418**** |
| **Humanity** | Love | .092 | -.216 | -.141 |
| | Kindness | .101 | -.164 | .144 |
| | Social Intelligence | .151 | **-.355*** | .007 |
| **Justice** | Teamwork | .048 | **-.438**** | -.286[†] |
| | Fairness | **.430**** | -.065 | .203 |
| | Leadership | **.421**** | -.128 | .176 |
| **Temperance** | Forgiveness | .024 | -.242 | .011 |
| | Humility | .005 | -.211 | -.080 |
| | Prudence | .151 | -.041 | .070 |
| | Self-Regulation | -.112 | **-.549***** | **-.507**** |
| **Transcendence** | Appreciation of Beauty and Excellence | .106 | -.165 | **.324*** |
| | Gratitude | -.116 | **-.622***** | **-.332*** |
| | Hope | -.195 | **-.541***** | **-.553***** |
| | Humor | .113 | -.211 | .114 |
| | Spirituality | -.190 | **-.403*** | -.183 |

Note.

[†] $p < .10$,

* $p < .05$,

** $p < .01$,

*** $p < .001$

participant did not identify with the feedback. Concerning the perceived importance of the topic of character strengths in therapy, the median rating (= 4.27; SD = 0.79) indicated a high level of perceived importance (scale range: min. = 1; max. = 5).

## Discussion

Generally, measuring psychopathology vs. character strengths via questionnaires is not associated with changes in psychological states, including mood, optimism, therapy motivation, and perceived stigma. However, there were small-sized time effects that emerged in the form of decreased negative affect and perceived stigma, as well as improved cognitive performance (measured by the TMT), in comparison to baseline, for both the pathology focus and the character strengths focus. The reduced completion times for the TMT tasks may potentially be attributed to a learning effect. The reduction in negative affect and perceived stigma in both conditions could be interpreted as the result of the comprehensive assessment of symptoms and character strengths within the scope of this study, which may have led the participants to feel more perceived and less discriminated. In addition or alternatively, anticipatory anxiety or

stress at the beginning of the study may have simply decreased over time (regression to the mean). As suggested by some studies, the phenomenon of initial elevation bias, whereby higher (especially negative) states are reported in initial assessments compared to later assessments, appears to be not uncommon in research utilizing subjective reports [24].

Character strengths interventions seem to be well accepted and are viewed as important by individuals diagnosed with psychosis. Self-regulation descriptively is the least, kindness the most strongly endorsed strength in the current sample. Kindness was also among the most endorsed strengths in a sample with first episode psychosis, as were honesty and fairness [9]. Correlations in the present study were most prominent with negative symptoms, hence the higher negative symptoms the less strongly various strengths are endorsed. This is consistent with cognitive theories of negative symptoms that state that a broadly negative self-image and negative (e.g., defeatist performance-) beliefs could contribute to amotivation, anhedonia, and avolition [25].

Addressing negative symptoms is highly important given the role for long-term functioning. This is supported by the aforementioned findings of Beck et al. [7], which suggest that motivational and attitudinal factors that are severely impaired in individuals with negative symptoms (e.g. amotivation, low effort, social withdrawal) have a significant impact on cognitive performance and the level of functioning. These motivational and attitudinal factors are in turn likely based upon dysfunctional beliefs, such as defeatist performance beliefs. Given these cognitive models of negative symptoms and functionality, it appears to be of great importance for the treatment of schizophrenia to inactivate and weaken dysfunctional beliefs and enhance positive attitudes and views of the self. Based on this principle, Aaron T. Beck and colleagues have developed a therapeutic approach, namely recovery-oriented cognitive therapy (CT-R) [26], which has been shown to be particularly promising for problems that are difficult to treat with medication, such as negative symptoms and poor functional outcomes. This is supported by the results of a randomized controlled trial conducted by Grant et al. [27], in which low-functioning patients with schizophrenia were randomly assigned to either receive CT-R or treatment as usual (TAU). The study found that the individuals who received CT-R showed significantly stronger improvements in global functioning and significantly stronger reductions in negative and positive symptoms compared to the TAU group. Moreover, a six-month follow-up assessment showed that the significant benefits of recovery-oriented cognitive therapy over standard treatment were maintained with comparable effect sizes [28]. The negative correlations found in this study between negative symptoms and character strengths could indicate that identifying and promoting character strengths in individuals with psychotic disorders may represent a fruitful therapeutic approach in the sense of recovery-oriented cognitive therapy, especially concerning negative symptoms.

Limitations of the present study are, for one, the sampling approach. Some individuals reported a first episode (20.5%), most reported multiple episodes (69.2%), and some only past episodes (10.3%). The effects of the intervention could differ by the degree of chronification and this should be further investigated in future studies with larger samples, whereby moderation effects could be examined. Another point of criticism is the reliance on an online format and hence self-reported diagnoses, lacking a differentiated formal diagnosis (e.g., using the SCID-5). However, applying a lie scale, and descriptively comparing symptom scores of the CAPE in the current sample to other samples, suggested plausible expressions of psychotic symptoms in the present sample. As a matter of fact, the study was initially planned to be conducted face-to-face which would have allowed for a diagnostic assessment, but was then conducted online due to strict COVID-19 policies in Germany during the planned recruitment period. A face-to-face format could have furthermore elicited stronger effects of the

manipulations, since a social component of the experimenter conducting the assessments and, perhaps, also giving feedback, could have had a much stronger impact upon the participants' psychological states and performance. Another reason for the manipulation not being successful might have been that individuals could have self-criticized internally for lacking the strength at question, albeit the general improvement over time in negative affect, perceived stigma, and cognitive performance somewhat contradicts this notion. Instead of relying on longer blocks of assessment, and to separate the manipulation from the rest of the study, it would have been beneficial to work with feedback on strengths vs. feedback on deficits or pathology. In addition, it is possible that attempting a psychological state change induction twice and in opposite hypothesized directions during one experiment (within sample) may have been too ambitious–hence replication studies may use longitudinal or between-subject designs. Finally, the focus of the study was on the effect of assessing strengths vs. deficits, and not the effect of character strength feedback per se. However, evaluating the effect of the strength feedback on psychological states might have been insightful beyond the assessment of overall perceived relevance of the topic and subjective benefit, and this should be implemented in future studies. The reported reliabilities of the scales, which refer to the sample of this study, were generally good ($\alpha > .80$), however three of the VIA-IS scales for the character strengths showed unsatisfactory reliability of a Cronbach's $\alpha < .60$ (i.e., *fairness*, $\alpha = .59$; *kindness*, $\alpha = .51$; *prudence*, $\alpha = .32$), which should be taken into account when interpreting results concerning these three subscales.

We conclude that positive psychology character strengths interventions are well accepted and perceived as relevant in a mixed sample of individuals with psychosis. Future studies should focus on the effects of strengths vs. deficit assessment and feedback, and use longitudinal, or between-subject approaches to study supposed changes in psychological states. Further research on the topic of character strengths interventions in psychosis seems worthwhile especially from the theoretical background of negative self-appraisal and defeatist beliefs being commonly related to negative symptoms and cognitive performance in psychosis, which play a major role in the quality of life and long-term functioning of affected individuals.

## Author Contributions

**Conceptualization:** Simone Kühn, Daniel Schöttle, Steffen Moritz, Jürgen Gallinat, Leonie Ascone.

**Data curation:** Aman Randhawa, Leonie Ascone.

**Formal analysis:** Aman Randhawa, Leonie Ascone.

**Investigation:** Aman Randhawa, Leonie Ascone.

**Methodology:** Simone Kühn, Daniel Schöttle, Steffen Moritz, Jürgen Gallinat, Leonie Ascone.

**Project administration:** Simone Kühn, Jürgen Gallinat, Leonie Ascone.

**Resources:** Simone Kühn, Daniel Schöttle, Steffen Moritz, Jürgen Gallinat, Leonie Ascone.

**Software:** Leonie Ascone.

**Supervision:** Simone Kühn, Daniel Schöttle, Steffen Moritz, Jürgen Gallinat, Leonie Ascone.

**Validation:** Steffen Moritz, Leonie Ascone.

**Visualization:** Aman Randhawa, Leonie Ascone.

**Writing – original draft:** Aman Randhawa, Leonie Ascone.

**Writing – review & editing:** Aman Randhawa, Simone Kühn, Daniel Schöttle, Steffen Moritz, Jürgen Gallinat, Leonie Ascone.

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
