## [Decision Letter · Decision Letter 0]

6 Mar 2023

PONE-D-22-26122The Effects of Assessing Character Strengths vs. Psychopathology on Mood, Hope, Perceived Stigma and Cognitive Performance in Individuals with PsychosisPLOS ONE

Dear Dr. Aman Randhawa

Thank you for submitting your manuscript to PLOS ONE. After careful consideration, we feel that it has merit but does not fully meet PLOS ONE’s publication criteria as it currently stands. Therefore, we invite you to submit a revised version of the manuscript that addresses the points raised during the review process. As you can see, each one of the reviewers raise issues that are complementary. Therefore, when revising your manuscript, please carefully considere and address the reviewers comments.  Based upon these reviews and also my own reading of your manuscript, I think that the results of your study needs to be discussed in a more substantial way. Specifically, you found negative associations between character strenghts (and virtues) and negative symptoms (also with some depressive symptoms). I considere that these results are important for the understanding of psychological features in individuals with psychosis, not only in descriptive terms but also are important to intervention. As a consequence, I invite you to help the readers to better understand these relations. 

We look forward to receiving your revised manuscript.

Kind regards,

Paulo Moreira, PhD

Academic Editor

PLOS ONE

Journal Requirements:

Reviewers' comments:

Reviewer's Responses to Questions

**Comments to the Author**

1. Is the manuscript technically sound, and do the data support the conclusions?

Reviewer #1: Yes

Reviewer #2: Yes

2. Has the statistical analysis been performed appropriately and rigorously? 

Reviewer #1: Yes

Reviewer #2: Yes

3. Have the authors made all data underlying the findings in their manuscript fully available?

Reviewer #1: Yes

Reviewer #2: Yes

4. Is the manuscript presented in an intelligible fashion and written in standard English?

Reviewer #1: Yes

Reviewer #2: Yes

5. Review Comments to the Author

Reviewer #1: The manuscript “The Effects of Assessing Character Strengths vs. Psychopathology on Mood, Hope, Perceived Stigma and Cognitive Performance in Individuals with Psychosis” provides an interesting multidisciplinary approach to better understanding the relevance of character strengths in individual with psychosis, as well as state effects of completing questionnaires. The method and analyses are straightforward, and the manuscript is overall well written. The introduction captures the central theoretical and empirical background, and limitations and implications are included in the discussion.

There are several comments that would help to further improve the manuscript:

1. There are a few linguistic corrections that should be made (e.g. “acceptance of focusing character strengths”, p. 9; “developed by Timo Gnambs for in Unipark”, p. 11)

2. In the study, the 120-version of the VIA-IS was used. The full version comprises 240 items, and it should hence be mentioned that the short version was used, and the appropriate citation and reference should be added.

3. Please report the reliabilities of each scale of the employed measures across the full sample. These could be summarised with range and median for the VIA-IS scales given the large number of scales.

4. In Table 2, it is not explained why certain values are in bold and what the asterisks indicate. Please add this information in the Note. Furthermore, as the correlations were across a small sample, Pearson correlations could be over- or underestimated due to outliers, even if each variable is approximately normal distributed. Hence, I would rather recommend conducting Spearman rank correlations throughout to avoid these biases.

Reviewer #2: Title: The Effects of Assessing Character Strengths vs. Psychopathology on Mood, Hope, Perceived Stigma and Cognitive Performance in Individuals with Psychosis.

Manuscript Number: PONE-D-22-26122.

Dear editor and authors.

Thanks for the opportunity to review this interesting paper.

The authors aimed to investigate whether assessments of psychopathology vs. character strengths were associated with differences on psychological states such as cognitive performance, state mood, optimism, therapy motivation or perceived stigma, in individuals with self-reported psychotic disorders.

In addition, they evaluated the acceptance and appraisal of a short online character-strength intervention, consisting of top-two strengths feedback, and to explore associations between character strengths and psychotic symptoms.

The study was conducted online and designed as randomized-controlled within-subject cross-over design. At the baseline, all participants (N = 39) were assessed by standardized measures on psychological states: cognition (TMT A/B), positive and negative affect, motivation for change/therapy, optimism, and self-stigma.

After the baseline assessment, the participants were randomized into two blocks: 1) the participants first filled out a psychopathology vs. character strengths assessment block. This was followed by repeated assessment of cognition and psychological states. 2) the respective other assessment block (character strengths vs. psychopathology) was presented, followed by a final assessment of cognition and psychological states.

Also, a brief intervention was applied. It consisted of targeted feedback regarding personalized character strengths, informing the participant about the definition and meaning of these strengths, and a short psychoeducation about the benefits of identifying and applying one’s strengths in everyday life.

A series of paired t-tests was calculated to compare whether there were differences in mood, optimism, therapy motivation, and perceived stigma:

a) between baseline vs. after the assessment block of character strengths

b) between baseline vs. after the assessment block of psychopathology

c) between the two assessment blocks of character strengths vs. psychopathology.

Moreover, correlation analyses were carried out to check associations between psychotic symptom domains and character strengths.

Finally, qualitative data (patient satisfaction) was evaluated descriptively.

Contrary to the author’s expectations, no differences between the psychological states after the pathology vs. character strengths assessment blocks were found.

However, some character strengths correlated negatively with positive (3 strengths) and depressive symptoms (3 strengths), but mainly with negative symptoms (12 of a total of 24 strengths), with medium to large effect sizes.

Furthermore, participants were satisfied with the brief intervention and rated a focus on personal strengths in psychotherapy as highly important.

Regardless the non-significant results about the main hypothesis, the authors concluded that the associations of character strengths with negative symptoms found in the study are important from the background of the cognitive model, which could be addressed in experimental research or clinical intervention studies targeting character strengths.

Results are discussed throughout the paper.

The paper’s objectives are interesting. The study is well-conducted and uses appropriated statistics. However, some issues could be listed:

The introduction is too short and could benefit to cite some other works on effects of attitudinal or motivational factors on neurocognitive performance. For example: Beck, A.T., Himelstein, R., Bredemeier, K., Silverstein, S.M., Grant, P. (2018). What accounts for poor functioning in people with schizophrenia: A re-evaluation of the contributions of neurocognitive v. attitudinal and motivational factors. Psychol Med. 48(16):2776-85.

Furthermore, recent development of Cognitive Therapy [Beck, A.T., Grant, P.M., Inverso, E., Brinen, A.P., & Perivoliotis, D. (2020). Recovery oriented cognitive therapy for serious mental health conditions. Guilford Press; Grant, P.M., Bredemeier, K., & Beck, A.T. (2017). Six-month follow-up of recovery-oriented cognitive therapy for low functioning individuals with schizophrenia. Psychiatric Services, 68, 997–1002] could be cited and discussed in the discussion section.

Moreover, the rationale and aims of the study should be present more clearly.

The methods section is well presented. However, the description of the procedure seems too wordy and difficult to follow. It could be described more clearly.

In the results section (Table 1) TMT, negative affect and stigma appears to be significantly different between baseline and the two assessment blocks. What does it mean? Authors should describe it and try to explain these differences.

Although the authors describe the main limitations of the study at the end of their manuscript, sample size and heterogeneity, sample recruitment (self-reported rather than formal diagnosis), and the decision not to measure the potential effect of the strengths character feedback intervention should be addressed more extensively.

6. PLOS authors have the option to publish the peer review history of their article (what does this mean?). If published, this will include your full peer review and any attached files.

Reviewer #1: No

Reviewer #2: **Yes: **Josep Andreu Pena Garijo

---

## [Author Response · Author response to Decision Letter 0]

12 Apr 2023

See "Response to Reviewers" file.

---

## [Decision Letter · Decision Letter 1]

26 Jun 2023

PONE-D-22-26122R1The effects of assessing character strengths vs. psychopathology on mood, hope, perceived stigma and cognitive performance in individuals with psychosisPLOS ONE

Dear Dr. Randhawa,

Thank you for submitting your manuscript to PLOS ONE. After careful consideration, we feel that it has merit but does not fully meet PLOS ONE’s publication criteria as it currently stands. Therefore, we invite you to submit a revised version of the manuscript that addresses the points raised during the review process.

We look forward to receiving your revised manuscript.

Kind regards,

Paulo Moreira, PhD

Academic Editor

PLOS ONE

Journal Requirements:

Reviewers' comments:

Reviewer's Responses to Questions

**Comments to the Author**

1. If the authors have adequately addressed your comments raised in a previous round of review and you feel that this manuscript is now acceptable for publication, you may indicate that here to bypass the “Comments to the Author” section, enter your conflict of interest statement in the “Confidential to Editor” section, and submit your "Accept" recommendation.

Reviewer #1: All comments have been addressed

Reviewer #2: All comments have been addressed

2. Is the manuscript technically sound, and do the data support the conclusions?

Reviewer #1: Yes

Reviewer #2: Yes

3. Has the statistical analysis been performed appropriately and rigorously? 

Reviewer #1: Yes

Reviewer #2: Yes

4. Have the authors made all data underlying the findings in their manuscript fully available?

Reviewer #1: Yes

Reviewer #2: Yes

5. Is the manuscript presented in an intelligible fashion and written in standard English?

Reviewer #1: Yes

Reviewer #2: Yes

6. Review Comments to the Author

Reviewer #1: The authors addressed in detail all the comments I raised on the original manuscript. I have a few additional remarks on the revised manuscript:

1. The newly added references for the VIA-IS short version both refer to the original, long versions of the questionnaire (with 240 items). The references for the short versions are Littman-Ovadia (2015) for the English version and Höfer, Hausler, Huber, Strecker, Renn and Höge (2020) for the German version.

2. The labelling of the character strengths in Figure 1, Table 2 and the text is inconsistent. It would be helpful to label them consistently throughout the manuscript.

Reviewer #2: (No Response)

7. PLOS authors have the option to publish the peer review history of their article (what does this mean?). If published, this will include your full peer review and any attached files.

Reviewer #1: No

Reviewer #2: **Yes: **Josep Andreu Pena Garijo

---

## [Author Response · Author response to Decision Letter 1]

19 Jul 2023

See "Response to Reviewers" file.

---

## [Editor Report · Decision Letter 2]

28 Jul 2023

The effects of assessing character strengths vs. psychopathology on mood, hope, perceived stigma and cognitive performance in individuals with psychosis

PONE-D-22-26122R2

Dear Dr. Randhawa,

We’re pleased to inform you that your manuscript has been judged scientifically suitable for publication and will be formally accepted for publication once it meets all outstanding technical requirements.

Kind regards,

Paulo Moreira, PhD

Academic Editor

PLOS ONE
---

## [Editor Report · Acceptance letter]

1 Aug 2023

PONE-D-22-26122R2 

The effects of assessing character strengths vs. psychopathology on mood, hope, perceived stigma and cognitive performance in individuals with psychosis 

Dear Dr. Randhawa:

I'm pleased to inform you that your manuscript has been deemed suitable for publication in PLOS ONE. Congratulations! Your manuscript is now with our production department. 

Kind regards, 

on behalf of

Professor Paulo Moreira 

Academic Editor

PLOS ONE